# The Role of Students’ Perceptions of Educators’ Communication Accommodative Behaviors in Classrooms in China

**DOI:** 10.3390/bs15040560

**Published:** 2025-04-21

**Authors:** Dan Ji, Howard Giles, Wei Hu

**Affiliations:** 1School of Humanity, Shanghai Jiao Tong University, Shanghai 200240, China; jidan@sjtu.edu.cn; 2School of International and Public Affairs, China Institute for Urban Governance, Shanghai Jiao Tong University, Shanghai 200240, China; 3Department of Communication, Santa Barbara, CA 93106-4020, USA; howiegiles@cox.net; 4School of Psychology, The University of Queensland, Brisbane, QLD 4072, Australia; 5School of Economics and Management, Tongji University, Shanghai 200092, China

**Keywords:** education, communication accommodation theory, communication satisfaction, teacher credibility, learning effectiveness

## Abstract

In China, educators are encouraged by administrators to assume instructional and language strategies to align with their students’ needs so as to enhance classroom communicative effectiveness, with students’ perceptions of teachers’ behavior being a salient factor in this process. This study, based on communication accommodation theory, examines how students’ perceptions of teachers’ behaviors influence reports of positive classroom outcomes. Using structural equation modeling, we analyzed responses from a sample of 422 university students in Shanghai. The results showed that the students’ perceptions of teachers’ communication accommodation behaviors, such as verbal and nonverbal tactics, teaching content, and emotional support behaviors, significantly and positively impacted students’ learning effectiveness, teacher credibility, and communication satisfaction. Furthermore, teacher credibility partially mediated the relationship between perceptions of accommodation and learner effectiveness. The findings offer practical insights for educators by suggesting that strategic adaptions of communication accommodations behaviors can promote students’ learning outcomes.

## 1. Introduction

Learning is an active, constructive process in which students gain tacit knowledge through their perceptions of the student–teacher relationship, which is important for educational outcomes. The teacher’s role is very influential in accounting for the variance in students’ academic achievement beyond the effects of particular other characteristics, such as principal and school characteristics ([14]). The extant literature has demonstrated that teachers’ communicative and supportive behavior is vital to promoting students’ engagement and satisfaction ([42]). More specifically, [44] ([44]) contended that teachers’ communicative behaviors were correlated with students’ perceptions of them which, in turn, could influence the latter’s learning outcomes (see also, [33]). In similar fashion, [13] ([13]) showed that teachers in language classes who used “simpler” grammatical structures and vocabulary based on their level of students’ capabilities facilitated learning. Not surprisingly, teachers with a more engaging communication style are often associated with enhanced educational effectiveness ([2]).

Teachers’ communication behaviors can be analyzed by communication accommodation theory (CAT), which serves as a valuable theoretical framework for understanding and enhancing communication dynamics. CAT emphasizes that individuals alter their communication strategies to achieve their desired interactional goals. This theory has been studied across disciplines, such as linguistics, sociology, sociolinguistics, and psychology ([13]), and has been even applied in biological contexts to examine communication in non-human species ([31]). However, its application in education settings remains underexplored, especially in China, where communication research and theory have never been associated with classroom investigations. Furthermore, prior research employing CAT in education has been dominated by qualitative research ([12]; [49]), whereas quantitative studies have provided additional valuable insights for they examine differences in levels or perceived levels of accommodative behaviors in different education contexts ([63]; [17]).

This study seeks to address these gaps by adopting structural equation modeling (SEM) to examine the relationship between Chinese students’ perceptions of teachers’ communication accommodation behaviors and their learning effectiveness. Additionally, teacher credibility and students’ communication satisfaction are proposed as potential mediating variables. Understanding these relationships may also assist teachers in creating more supportive and effective learning environments by identifying what works in the classroom.

Our research makes several contributions to the literature. First, it enriches the existing communication theories in educational settings by providing empirical evidence for the relevance of communication accommodation theory in the Chinese international education context because most of these studies have been conducted in Europe ([62]). Second, our findings offer critical insights into developing teachers’ communication competencies, underscoring the link between the perceptions of teachers’ communication accommodation behavior and learning effectiveness. Third, this study reveals the critical influence of students’ perceptions of communication satisfaction and teacher credibility in facilitating effective classroom communication, thereby offering practical implications for instructional communication strategies.

## 2. Literature Review and Theoretical Backdrop

In an arguably more theoretical vein, [17] ([17]) advocated that communication accommodation theory (CAT) could enhance our understanding of how language is enacted and received in classroom and educational training settings. CAT serves to describe and explain how and why people adjust their communication styles, toward or away from others, to take into account their interlocutors’ characteristics for various social functions, such as gaining approval, respect, and social identity maintenance. The theory also relates to how and why recipients subjectively and objectively respond to such moves and how these are processed by them. Objective accommodation refers to the linguistic and behavioral adjustments made by communicators, such as modifying speech patterns, using inclusive language, or adjusting tone and formality. However, subjective accommodation is the recipient’s perception and interpretation of these communicative efforts, which can be influenced by personal experiences, expectations, and cultural backgrounds. Importantly, these two dimensions do not always align—what one person intends as an accommodating gesture may not be perceived as such by another. Understanding this distinction allows people to better assess how their communication is received and to refine their strategies accordingly.

CAT has received substantial support in empirical studies across many nations and languages, social groups, and applied social contexts (e.g., [19]); for a recent review of the 50 years of CAT work to date, see [20] ([20], [21]).

In the educational context, CAT has the potential to inform teachers on how to effectively adjust their messages and communication patterns to improve student comprehension and classroom involvement, particularly (thus far) in second language learning settings ([54]; [62]). Research has indicated that teachers’ communication accommodative behaviors not only assist in building rapport but also correlate with students’ learning outcomes, including engagement, participation willingness, and classroom interaction ([39]; [40]; [46]). As above, [17] ([17]) suggested that promoting CAT practices may encourage teachers to refine their instructional approaches

However, most studies on this topic have primarily evaluated learning outcomes from teachers’ perspectives ([6]; see, however, [1]), with few considering students’ viewpoints ([44]; [63]). Consequently, our study focused on investigating how students perceive their teachers’ communication accommodative behaviors and the impact of these on learning outcomes, such as communication satisfaction, teacher credibility, and learning effectiveness.

Teachers’ communication accommodative behaviors include nonverbal behavior, verbal behavior, teaching content, and emotional support ([17]; [59]). Nonverbal behavior refers to the communication information through means such as facial expressions, eye contact, body posture, and gestures without the use of language. Verbal behavior involves the use of language (both spoken and written) to convey information. Teaching content refers to the information that teachers convey in the classroom are simplified for students’ understanding. Emotional support refers to the emotional care and psychological support that teachers provide in the classroom to create a positive learning atmosphere. These dimensions together constitute the overall framework of teachers’ communication accommodation behaviors, reflecting how teachers adjust their behaviors in various ways to meet students’ needs and thereby influence learning effectiveness.

Collecting data from students is not only cost-effective and efficient but also provides a more candid assessment of the classroom environment ([5]).

Among the various indicators related to classroom outcomes, we emphasized learning effectiveness—a crucial aspect of education—because it directly reflects the extent to which educational objectives are achieved. Learning effectiveness indicates how well learning objectives are achieved and how well learners’ needs are satisfied ([36]; [61]). In this way, our study—which is the first of its kind invoking CAT in China—focuses on exploring the roles of communication satisfaction and teacher credibility in the relationship between students’ perceptions of teachers’ communication accommodation behaviors and learning effectiveness (see [56]). Communication satisfaction (see [29]; [37]) stemming from positive communication experiences with instructors is known to facilitate positive educational outcomes ([23]). Instructor credibility is also influenced by accommodative behaviors, such as students’ perceptions of instructors using positive slang ([41]).

Furthermore, when teachers are construed as more credible, students are more engaged in classroom activities, more motivated to learn ([67], [66]), and more willing to attend classes ([50]). In tandem, they themselves experience increased communication confidence ([45]) and achieve improved learning outcomes ([47]). In light of this, the following research hypothesis was posed:

**H1:** *Perceptions of teachers’ communication accommodative behaviors will have a positive impact on students: (a) students’ communication satisfaction, (b) teacher credibility, and (c) learning effectiveness*.

However, [20] ([20]) discussed CAT research, demonstrating that while accommodative and nonaccommodative behaviors could have direct effects on others’ cognitions, affect, and behaviors (as in H1), accommodative moves sometimes acted more indirectly, triggering mediating mechanisms (e.g., trust) which, in turn, shaped or were responsible for pro-social behaviors.

Empirical studies across general, language, and communication education disciplines have underscored a robust positive correlation between teacher credibility and student outcomes, including the willingness to attend classes ([50]), engagement ([66]), and foreign language achievement ([54]).

Teacher credibility is a well-examined construct that plays a fundamental role in understanding learning effectiveness. Higher teacher credibility correlates with better learning outcomes ([56]). When students perceive their teachers as credible, they demonstrate greater motivation to learn ([66]), enhanced communication confidence ([45]), and improved learning outcomes ([47]).

Extant findings also support that teacher credibility plays a key mediating role in facilitating teacher–student interactions, and ultimately, classroom learning ([15]). [57] ([57]) found that teacher credibility mediated the relationship between teacher communication messages (e.g., clarity, confirmation, and nonverbal immediacy) and student learning outcomes (e.g., learner empowerment and affective learning).

Several other arguments also support the mediating role of communication satisfaction in teacher behavior and learning outcomes ([3]; [35]). [58] ([58]) suggested that communicating helps students better understand teachers’ expectations and then perform better.

Hence, and with the possibility that communication satisfaction could fulfill a similar role, we posited the next hypothesis:

**H2:** *Students’ communication satisfaction and teacher credibility have mediating effects on the link between perceptions of teachers’ communication accommodative behaviors and student learning effectiveness*.

The research model is shown in Figure 1.

Related studies have found that demographic heterogeneity can influence students’ perceptions. [38] ([38]) pointed out that learners’ age affects students’ perceptions of teachers’ adaptive behaviors. People over the age of 25 are more likely to notice adaptive behaviors. At the same time, women are more sensitive than men and are more likely to perceive teachers’ respectful attitudes and give higher evaluations. [48] ([48]) found significant differences in perceptions of the teachers, according to the students’ gender, class membership, and transition pathway. Given our relatively large sample with its inherent demographic heterogeneity (see Table 1 below), we also posed a background research question allied to the above:

**RQ**: In what ways, if at all, do individual differences, such as age, gender, educational level, and nation, influence student perceptions of teachers’ communication accommodative behaviors and credibility as well as communication satisfaction and learning effectiveness?

## 3. Method

### 3.1. Sampling and Data Collection

This study utilized a quantitative approach within the positivist paradigm. The study’s target population comprised Chinese and international students studying from multiple universities in Shanghai, including both public and private institutions. The sample included students from various departments, such as communication, business, and engineering, to ensure a broad representation. The final set of documents, including cover letter, and English and Mandarin questionnaire versions, was sent to respondents; the latter was included for the convenience of international students whose Mandarin proficiencies were not adequate for the task at hand. We used a rigorous back-translation process, where the questionnaire was translated from English to Mandarin and then back-translated to English by a different translator. Both versions were compared, and discrepancies were resolved through discussion and consultation with bilingual experts. A total of 500 surveys was distributed online through Wenjuanxing (https://www.wjx.cn/), a popular survey website with 33.24 million users in China. To ensure the integrity and reliability of the data, this study applied strict criteria to identify and exclude invalid questionnaires. Overall, 78 samples were excluded because (1) a large number of responses were missing; (2) the same option was chosen throughout the survey; (3) or the response time was too short without in-depth thinking. After applying the above exclusion criteria, a total of 422 valid responses remained, ensuring that the analysis of this study was based on high-quality data that could accurately reflect the students’ perspectives.

The dataset comprised 422 valid samples after data cleaning, resulting in an effective response rate of 84.4%. The total sample included 213 males (50.5%) and 209 females (49.5%), see Table 1 for more details.

### 3.2. Instructions

The research instruments, as introduced above, included four fundamental constructs: the students’ perceptions of: (1) teachers’ communication accommodative behaviors; (2) communication satisfaction; (3) teacher credibility; and (4) learning effectiveness. Each measurement item was adapted from prior studies. Participants reflected on their current teachers’ perceived communication accommodative behaviors (PCAB), using a scale developed by [17] ([17]) and [59] ([59]). This included four sub-dimensions: (1) nonverbal accommodation (NB), (2) verbal accommodation (VB), (3) teaching content (TC), and (4) emotional support behavior (ES). Communication satisfaction (CS) was measured by the 5-item scale developed by [23] ([23]). Teacher credibility (CR) was a revised version of [43] ([43]). Learning effectiveness (LE) was measured by the 4-item scale developed by [4] ([4]). A total of 39 items across the four constructs were obtained, as shown in Table 2.

More specifically and in line with [17] ([17]), participants were asked to reflect on their perceptions of their current teachers’ accommodation behaviors during the final two weeks of the semester using a 7-point scale ranging from “An Inappropriate Amount” (1) to “An Appropriate Amount” (7), as well as their communication satisfaction, the teacher credibility, and learning effectiveness using a Likert-type scale ranging from “Strongly Disagree” (1) to “Strongly Agree” (7).

Before the main data collection, a pilot study (N = 85) was conducted at Shanghai Jiao Tong University to assess the instrument’s clarity and face validity. After a reliability analysis of the items of responses, one item (VB4) with a low item-total correlation was deleted.

## 4. Data Analysis and Results

### 4.1. Common Method Bias

Before the model assessment, a priori and post hoc procedures were taken to address common method bias. Common method bias is problematic when a single latent variable accounts for the majority of the explained variance ([52]). The procedural remedies comprised piloting the survey instrument, including both the content and presentation. Harman’s single-factor test was used to assess the effect of common method bias. Harman’s single-factor test in SPSS Statistics 27 produced a result of 33% as the maximum variance explained by a single factor, which was lower than the threshold of 50% ([52]). Therefore, common method bias was considered not an issue in this study.

Following recommendations by [24] ([24]), the data were analyzed and interpreted in two stages, namely assessments of the measurement and the structural models.

### 4.2. Demographic Characteristics Difference in the Study Variables

Although not the main focus of the study, we began by addressing the RQ by computing descriptive statistics (means, SDs, and frequencies) to examine participants’ characteristics and the study variables. As above, there were the students’ perceptions of: (1) teachers’ communication accommodation behavior; (2) communication satisfaction; (3) teacher credibility; and (4) learning effectiveness. Inferential statistical analyses (*t*-test and one-way analyses of variance) were conducted to examine group differences and relationships among the study variables. All statistical analyses were conducted using the Statistical Package for the Social Sciences version 26.0. However, there were no significant differences among these variables, except communication satisfaction across different grade levels. To address the imbalanced sample sizes concerns on nationalities, we combined the smaller groups (“Japan”, “Korea”, “UK”, and “US”) into a collective “non-Chinese” category. This restructuring allowed us to perform a more robust comparison between Chinese and non-Chinese students, mitigating potential statistical biases from the previously imbalanced samples. The results presented in Table 3 show that there were no significant differences among all variables between Chinese and non-Chinese students. This indicates that nationality did not appear to significantly influence the measured constructs in this study. The other results were shown in Table 4, Table 5 and Table 6.

In sum, RQ did not present data or caveats to our main focus (albeit having implications for further investigation); we then moved to analyses relating to our two main hypotheses.

### 4.3. Data Analysis

This study employed structural equation modeling (SEM) analysis to evaluate the quality of the measurement tool and to test the proposed hypotheses using Smart PLS 3.0 software.

The analysis followed a two-stage approach, consisting of a measurement model and a structural model in line with ([53]). The measurement model assessed construct reliability, validity, indicator reliability, convergent validity, and discriminant validity. The structural model estimated the path and their significance levels. Smart PLS 3.27 software was utilized for structural equation modeling ([55]).

#### 4.3.1. Measurement Model

To confirm the reliability and validity of constructs, we first conducted a confirmatory factor analysis (CFA) to determine the measurement model’s fitness using covariance-based SEM (CB-SEM). According to [25] ([25]), we assessed the fit indices, including Chi-square = 919.84 and degrees of freedom (df) = 584, Chi-square/df = 1.58, Goodness-of-Fit Index (GFI) = 0.89, Comparative Fit Index (CFI) = 0.97, Normed Fit Index (NFI) = 0.93, Tucker–Lewis Index (TLI) = 0.97 and Root-Mean-Square Error of Approximation (RMSEA) = 0.04. All the indices were found to be within the acceptable range ([10]), except GFI, which met the 0.80 acceptable threshold, suggesting a reasonable fit ([11]).

We then assessed reliability, convergent validity, and discriminant validity. Reliability was determined by examining the internal consistency of the scales using Cronbach’s alpha, where a value above 0.70 is regarded as statistically reliable. As Table 3 shows, Cronbach’s alpha values for all constructs met this reliability criterion.

Convergent validity, which assesses whether different items measure the same construct, was evaluated through factor loadings, composite reliability (CR), and average variance extracted (AVE). Table 7 shows that almost all factor loadings exceeded the minimum requirement of 0.70. Due to insufficient loadings, three items were excluded following [32] ([32]).

Furthermore, composite reliability and the AVE for each construct exceeded the minimum values of 0.70 and 0.50, confirming that convergent validity was established ([16]).

Convergent validity measures whether items can effectively reflect their corresponding factor, whereas discriminant validity measures whether two constructs are statistically different. To evaluate discriminant validity among constructs, this research applied the Heterotrait–Monotrait (HTMT) ratio as suggested by ([30]). The HTMT scores for all constructs did not violate the threshold value of 0.90, thus confirming discriminant validity, as presented in Table 8.

#### 4.3.2. Structural Model

To assess collinearity in the structural model, as recommended by [28] ([28]), the presence of multicollinearity was examined using the variance inflation factor (VIF). A VIF value of 5 or above indicates the presence of multicollinearity. In this study, the VIF values were around five or below, indicating no issue with multicollinearity.

The structural models for this research are presented in Figure 2, where R2 represents the value for each endogenous and predicted latent variable. Previous studies have established threshold values of R2, with 0.75, 0.50, and 0.25 representing substantial, moderate, and weak relationships, respectively ([28]; [30]). For the dependent variable, learning effectiveness (LE), the R2 value was 0.314, indicating that it explained 31.4% of the variance in learning effectiveness.

According to [8] ([8]) criterion, the f2 values of 0.02, 0.15, and 0.35, respectively, represent small, medium, and large effects of the exogenous latent variable. The f2 values for CR → LE and CS → LE were 0.137 and 0.029, indicating small to large effect sizes. Furthermore, the predictive relevance of Stone–Geisser’s (Q2) was investigated using the blindfolding procedure in PLS-SEM. Q2 values greater than zero indicate that the exogenous variable has predictive relevance for the endogenous variable in the structural or inner model. In this study, the Q2 values for LE, CS, and CR were 0.264, 0.086 and 0.130, indicating the model’s predictive relevance ([26]).

Additionally, the standardized root means squared residual (SRMR), which indicates the approximate fit of the model was assessed. SRMR measures the difference between the observed correlation and the model-implied correlation matrix ([25]). A model is viewed as a good fit if it has an SRMR value less than or equal to 0.08 ([34]). The SRMR value of this study was 0.058, confirming that our model meets the criteria for fit based on its SRMR value.

According to [24] ([24]), after establishing an adequate research model, hypothesis testing can be carried out. In this study, the bootstrapping method was used to determine the statistical significance of the path coefficient and to calculate the t-values. This method aims to address the issue of abnormal research data.

##### Direct Effect

The hypothesis is accepted if the significance level is below 0.05 or the t-value exceeds the critical value ([24]). The results are shown in Table 9 and Figure 2. The findings show that all hypotheses produced positive and significant results, and the results presented in Table 9 and Figure 2 show that the variable of perceptions of teachers’ communication accommodative behaviors had a positive impact on students’ learning outcomings; therefore, H1 was supported.

##### Mediation Effects

The study followed guidelines provided by [24] ([24]) to test for mediation. Path analysis was conducted using the bootstrapping approach for 5000 samples computed at 95% confidence. Table 8 represent the indirect effects of communication satisfaction and teacher credibility on the relationships between the four components of the students’ perceptions of teachers’ accommodation behaviors and their learning effectiveness.

The Variance Accounted For (VAF) metric categorizes the mediating variables as full or partial mediation. Following the guidelines of [27] ([27]), a VAF value exceeding 80% indicates full mediation, a VAF value between 20% and 80% indicates partial mediation, and a VAF value below 20% indicates no mediation. The VAF value is calculated using the formula: VAF = Indirect Effect/(Direct Effect + Indirect Effect). Based on this, the results in Table 10 show that teacher credibility assumed a partial mediating role because all of its VAF value were between 20% and 80%. In contrast, communication satisfaction did not mediate the relationship between verbal accommodation and learning effectiveness nor teaching content and learning effectiveness. The direct impact of verbal and teaching content accommodation on learning effectiveness may be strong enough to overshadow the mediating role of communication satisfaction. This could occur when the accommodation behaviors are particularly effective or when other factors, such as student motivation or prior knowledge, play a more significant role in learning outcomes. And the mediating role of communication satisfaction may be influenced by contextual factors, such as the classroom environment, and individual student characteristics, et al. But overall, the mediating effect existed. Thus, H2 was partially supported.

## 5. Discussion

This study, drawing on CAT, provided evidence indicating how students’ perceptions of teachers’ communication accommodation behaviors currently affected their students’ reported levels of communication satisfaction, teacher credibility, and learning effectiveness, with the mediating role of teacher credibility. This work adds to the theoretical framework of CAT by empirically demonstrating the consistent attributional impacts of ver*y different forms* of teachers’ communication accommodative behaviors (i.e., nonverbal, verbal, teaching content, and emotional support) can have on students’ learning effectiveness, which aligns with the broader literature on communication competence, which emphasizes the importance of accommodative communication strategies in facilitating effective interactions ([51]). Typically, CAT studies focus on one or two behaviors (e.g., speech rate, pitch, language switching). It is important to point out that our study examined the impact of communicative accommodative behaviors in the context of international education in Shanghai, where differences in cultural and educational systems are apparent. This heterogeneity could have potentially incurred what might have been considered detrimental variance to discovering common patterns of findings. However, the robust trends emerging from the study do offer some cautious credence to the cross-cultural generalizability of CAT.

CAT has been conceptualized, historically, as involving different foci and stages ([20]). Relevant to the current study, Stage 6—referencing the first six benchmark stages of CAT’s evolution—emerged when researchers began examining the ways in which the effects of accommodative and nonaccommodative moves were mediated by certain indirect or intervening factors, such as trust or threat (e.g., [7]). Our investigation adds to the growth of work in this stage by showing how communication accommodative phenomena function *within the classroom* to influence the pivotal cognitive and affective elements of teacher credibility which, in turn, facilitate and transform educational outcomes. In parallel, CAT’s foundations and essence have been regularly revised in a set of CAT Principles (for 11 of these, see [20]). Some of these could be modestly embellished to concretely appeal to *positive educational* outcomes while also acknowledging less favorable ones when teacher *non*accommodations are apparent (see [17]). Clearly, the latter in themselves could not only engender an unfavorable educational climate but could foster, reciprocally and cyclically, student nonaccommodations in turn.

The findings of this study reveal significant differences in how students of various nationalities assess learning effectiveness. Notably, Chinese college students rated learning effectiveness the highest, followed by international students, with British and South Korean students ranking next. These differences suggest that students from diverse cultural backgrounds perceived the teaching behaviors differently. To foster inclusivity, universities should implement accommodative strategies such as cultural sensitivity training for educators and customized evaluation criteria that reflect diverse academic backgrounds.

We know that eye contact, using body language, using inclusive language, expressing empathy, acknowledging challenges, and so on will have an impact on the learning effect. Our findings also offer pragmatic insights for educators who should be encouraged to adopt communication accommodation strategies—as well as other so-called discourse management strategies not studied herein (see, for example, [9]; [65])—to influence student learning with a positive impact. In this sense, it is recommended that educators foster positive relationships with students not only through consistent verbal and nonverbal behaviors, but also through implementing sensitive communication accommodation strategies to the varying needs of different students to enhance their educational experiences. In order to augment students’ learning, educators should employ specific accommodation behaviors in the classroom that include, but are not limited to, nonverbal cues like smiling and maintaining eye contact, using gestures for emphasis, and being mindful of classroom positioning (e.g., strategically moving around the classroom, being attentive to seating arrangements, and adjusting seating when necessary). Relatedly, it is crucial to provide emotional support to students, including monitoring their engagement in the classroom, offering prompt feedback, and demonstrating genuine concern for their academic experiences. These practices are essential for creating an environment conducive to learning and academic success. Some effective teaching behaviors are similar to those suggested by previous researchers ([42]; [2]), but this study systematically verifies the impact of accommodative behaviors in the classroom from the perspective of CAT.

This study inevitably possesses limitations which should be addressed in subsequent research endeavors. The participants were confined to college students in Shanghai, and future work should reach out elsewhere to more expansive samples and diverse student and teacher demographics, taking into account their documented bi- or multilingual skills, and extending this to multiple nations for international comparative analyses. In addition, the sample could be expanded so as to inclusively study the effect of teaching across various academic disciplines (e.g., arts and sciences) and lesson types (e.g., writing and speaking).

We acknowledge that factors such as instructor gender, cultural background, teaching experience, class size, frequency of meetings, and course subject matter can influence students’ perceptions of communication accommodation. While we did not control for these variables in our study, we recognize their potential impact and suggest them as areas for future research.

The methodology for data collection herein was restricted to a questionnaire survey (including many questions), which raises the specter of self-report bias and participant fatigue, which may have influenced the reliability of responses. Critically, future studies should incorporate a range of qualitative methods, including conversational and discourse analyses, to examine, more naturalistically, actual and ongoing classroom practices, as well as include more objective measures of learning outcomes (see [18]). Finally, it is important to stress that our CAT-like inclinations did not favor theoretical hegemony as we fully subscribed to the integration of this theory with other models, such as communication theories of identity (e.g., [60]; [64]) for the benefit of intellectual advance me; in this regard, see many of the contributions in [22] ([22]).

## 6. Conclusions

In sum, this study adds to the ever-growing literature on CAT by clearly revealing, in a Chinese educational context, positive associations between students’ perceptions of teachers’ various communication accommodation behaviors with several factors, namely teacher credibility, communication satisfaction, and students’ reported learning effectiveness. Not only do these findings have applied implications for teachers’ discourse in the classroom, but it also opens up a dire need for more programmatic research across different populations and cultures, analytically taking into account the many different demographic (and other) characteristics of both students *and* teachers.

## Figures and Tables

**Figure 1 behavsci-15-00560-f001:**
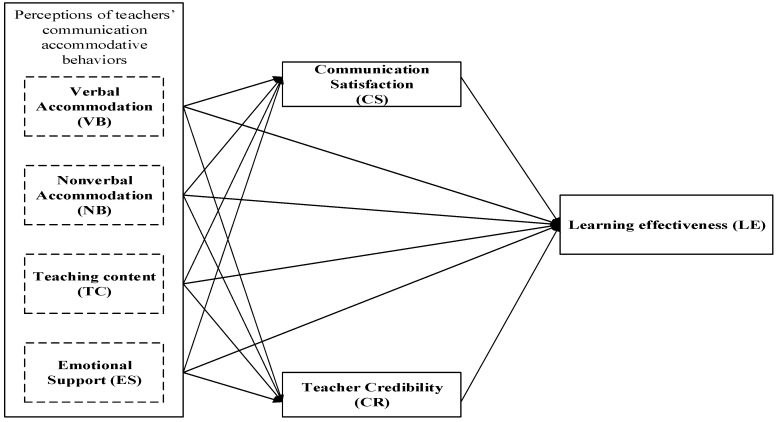
The conceptual model.

**Figure 2 behavsci-15-00560-f002:**
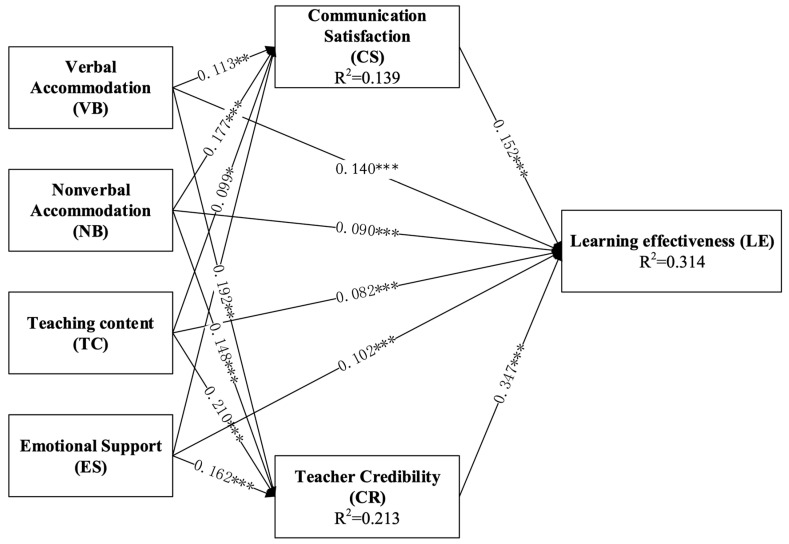
Structural model results (SEM) (* *p* < 0.05, ** *p* < 0.01, *** *p* < 0.001).

**Table 1 behavsci-15-00560-t001:** Demographic information of the respondents.

Demographic Variables	Item	Frequency	Percentage (%)
**Gender**	male	213	50.5
	female	209	49.5
**Age**	18–23	260	61.6
	24–29	134	31.8
	30–35	14	3.3
	36–40	9	2.1
	≥41	5	1.2
**Education level**	Freshman	68	16.1
	Sophomore	55	13
	Junior	68	16.1
	Senior	55	13
	Graduate student	34	8.1
	Postgraduate student	79	18.7
**Nation**	China	320	75.8
	Australia	1	0.2
	French	1	0.2
	Japan	20	4.7
	Korea	34	8.1
	UK	27	6.4
	Canada	1	0.2
	America	9	2.1
	Thailand	1	0.2
	Ukraine	1	0.2
	Other	6	1.4
	Malaya	1	0.2
**Number of responses**		422	100%

**Table 2 behavsci-15-00560-t002:** Survey items.

Items		Source
Student Perception of Teacher Accommodation Communication Behavior (PACB)		
Nonverbal behavior (NB)	1. Made eye contact with me 2. Smiled at me 3. Showed enthusiasm 4. Used gestures to emphasize points 5. Moved around the classroom when speaking	[17] ([17]) and [59] ([59])
Verbal behavior (VB)	1. Concentrated on articulating words for clarity 2. Tried to use simple language 3. Made an effort to pronounce words correctly 4. Used slang that I would use	
Teaching content (TC)	1. Provided feedback to me 2. Incorporated examples to make course content relevant 3. Explained course content thoroughly 4. Repeated his/her ideas to help me understand	
Emotional support (ES)	1. Provided emotional support 2. Made me feel comfortable 3. Was concerned about my success in the class 4. Was responsive to my needs 5. Empathized with me	
**Students’ communication satisfaction (CS)**		
	1. My communication with my teacher feels satisfying. 2. My teacher fulfills my expectations when I talk to him/her. 3. My conversations with my teacher are worthwhile. 4. My teacher genuinely listens to me when I talk. 5. My teacher tends to dominate our conversations and not allow me to get my point across.	[23] ([23]) and [29] ([29])
**Teacher’s credibility (CR)**		
Competence	1. I think he/she is competent to teach this course. 2. I think the teacher has received professional training and is highly professional. 3. I think he/she is full of wisdom. 4. I think he/she lacks professional knowledge.	[43] ([43])
Goodwill	1. I think he/she is concerned about students. 2. I think he/she attached importance to students’ interest in learning. 3. I think he/she is self-centered. 4. I think he/she can understand students.	
Trustworthiness	1. I think he/she is honest and trustworthy. 2. I think he/she is trustworthy. 3. I think he/she is honest. 4. I think he/she abides by ethics.	
**Learning effectiveness (LE)**		
	1. Instructor’s teaching methods are flexible and diverse and can adapt to individual differences. 2. The instructor explained this clearly in class. 3. The instructor can give timely feedback on students’ questions. 4. The instructor was open to students’ views.	[4] ([4])

**Table 3 behavsci-15-00560-t003:** Independent samples *t*-test for the impact of students’ nationality on different variables.

	Nationality	N	Mean	Standard Error	t	*p*
Nonverbal behavior	Chinese	320	4.79	1.82	0.018	>0.05
	non-Chinese	102	4.15	1.81		
Verbal behavior	Chinese	320	4.07	2.01	0.455	>0.05
	non-Chinese	102	4.21	1.94		
Teaching content	Chinese	320	4.11	1.54	0.00	>0.05
	non-Chinese	102	4.15	1.67		
Emotional support	Chinese	320	4.47	1.81	−0.134	>0.05
	non-Chinese	102	4.22	1.99		
Communication satisfaction	Chinese	320	4.63	1.29	1.646	>0.05
	non-Chinese	102	4.33	1.37		
Teacher’s credibility	Chinese	320	4.72	1.31	1.813	>0.05
	non-Chinese	102	4.43	1.27		
Learning effectiveness	Chinese	320	4.43	1.37	1.180	>0.05
	non-Chinese	102	4.17	1.43		

**Table 4 behavsci-15-00560-t004:** Independent samples *t*-test for the impact of students’ gender on different variables.

	Gender	N	Mean	Standard Error	t	*p*
Nonverbal behavior	F	209	4.34	1.72	1.74	>0.05
	M	213	4.12	1.90		
Verbal behavior	F	209	4.01	2.02	−0.86	>0.05
	M	213	4.08	1.96		
Teaching content	F	209	4.06	1.63	−0.65	>0.05
	M	213	4.15	1.52		
Emotional support	F	209	4.05	1.87	−0.16	>0.05
	M	213	4.38	1.84		
Communication satisfaction	F	209	4.50	1.33	−1.04	>0.05
	M	213	4.63	1.28		
Teacher’s credibility	F	209	4.59	1.31	−1.02	>0.05
	M	213	4.72	1.30		
Learning effectiveness	F	209	4.38	1.42	0.05	>0.05
	M	213	4.38	1.36		

**Table 5 behavsci-15-00560-t005:** One-way ANOVA results for age.

	Age	N	Mean	SD	F	*p*
Communication satisfaction	18–23	260	4.62	1.30	0.70	>0.05
	24–30	134	4.53	1.34		
	31–35	14	4.07	0.99		
	36–40	9	4.36	1.45		
	>41	5	4.64	0.92		
Teacher’s credibility	18–23	260	4.65	1.34	0.98	>0.05
	24–30	134	4.75	1.25		
	31–35	14	4.21	1.09		
	36–40	9	4.10	1.29		
	>41	5	4.63	1.42		
Learning effectiveness	18–23	260	4.42	1.39	1.32	>0.05
	24–30	134	4.34	1.41		
	31–35	14	4.89	1.28		
	36–40	9	4.08	1.24		
	>41	5	5.41	1.02		
Nonverbal behavior	18–23	260	4.78	1.87	0.43	>0.05
	24–30	134	4.89	1.70		
	31–35	14	4.50	1.54		
	36–40	9	4.53	2.26		
	>41	5	4.08	2.46		
Teaching content	18–23	260	4.01	1.59	0.84	>0.05
	24–30	134	4.29	1.52		
	31–35	14	4.91	1.48		
	36–40	9	4.14	2.18		
	>41	5	4.55	0.69		
Emotional support	18–23	260	4.70	1.90	0.35	>0.05
	24–30	134	4.91	1.79		
	31–35	14	4.74	1.51		
	36–40	9	4.49	2.40		
	>41	5	5.08	0.97		
Verbal behavior	18–23	260	4.05	2.04	1.24	>0.05
	24–30	134	5.79	1.96		
	31–35	14	4.88	0.78		
	36–40	9	4.52	2.38		
	>41	5	4.20	1.07		

**Table 6 behavsci-15-00560-t006:** One-way ANOVA results for grades.

	Grades	N	Mean	SD	F	*p*	Post Hoc Test
Communication satisfaction	Freshman	68	4.16	1.36	2.62	<0.05	6 < 1 < 2 1 < 4 1 < 5
	Sophomore	55	4.91	1.26			
	Junior	34	4.46	1.31			
	Senior	79	4.68	1.30			
	Graduate student	135	4.66	1.24			
	Postgraduate student	51	4.40	1.36			
Teacher’s credibility	Freshman	68	4.47	1.25	1.45	>0.05	
	Sophomore	55	4.59	1.48			
	Junior	34	4.38	1.26			
	Senior	79	4.66	1.35			
	Graduate student	135	4.88	1.25			
	Postgraduate student	51	4.59	1.26			
Learning effectiveness	Freshman	68	4.24	1.35	0.46	>0.05	
	Sophomore	55	4.43	1.39			
	Junior	34	4.21	1.28			
	Senior	79	4.34	1.43			
	Graduate student	135	4.50	1.42			
	Postgraduate student	51	4.37	1.39			
Nonverbal behavior	Freshman	68	4.79	1.86	2.05	>0.05	
	Sophomore	55	4.06	1.79			
	Junior	34	4.95	2.29			
	Senior	79	4.78	1.85			
	Graduate student	135	4.96	1.69			
	Postgraduate student	51	4.60	1.62			
Teaching content	Freshman	68	4.91	1.70	1.36	>0.05	
	Sophomore	55	4.81	1.44			
	Junior	34	4.82	1.67			
	Senior	79	5.18	1.64			
	Graduate student	135	5.27	1.50			
	Postgraduate student	51	5.13	1.54			
Emotional support	Freshman	68	5.74	2.01	0.31	>0.05	
	Sophomore	55	5.64	1.68			
	Junior	34	4.67	1.92			
	Senior	79	4.65	2.09			
	Graduate student	135	4.84	1.76			
	Postgraduate student	51	4.98	1.68			
Verbal behavior	Freshman	68	4.35	1.67	0.74	>0.05	
	Sophomore	55	4.90	2.20			
	Junior	34	4.67	2.40			
	Senior	79	4.83	2.24			
	Graduate student	135	4.60	1.83			
	Postgraduate student	51	4.98	1.87			

Note: In the post hoc test, “1” represents freshman, “2” represents sophomore, “3” represents junior, “4” represents senior, “5” represents graduate student, and “6” represents postgraduate student.

**Table 7 behavsci-15-00560-t007:** Assessment of construct reliability and convergent validity.

	Items	Factor Loading	AVE (>0.5)	Composite Reliability (>0.6)	Cronbach’s Alpha (>0.7)
Nonverbal behavior (NB)	NB1	0.894	0.802	0.953	0.939
	NB2	0.901			
	NB3	0.913			
	NB4	0.910			
	NB5	0.857			
Verbal behavior (VB)	VB1	0.930	0.876	0.955	0.930
	VB2	0.936			
	VB3	0.943			
Teaching content (TC)	TC1	0.887	0.791	0.938	0.912
	TC2	0.909			
	TC3	0.887			
Emotional support (ES)	ES1	0.907	0.818	0.818	0.945
	ES2	0.917			
	ES3	0.899			
	ES4	0.891			
Communication satisfaction (CS)	CS1	0.873	0.648	0.900	0.858
	CS2	0.853			
	CS3	0.850			
	CS4	0.854			
	CS5	0.545			
Teacher’s credibility (CR)	CR1	0.843	0.628	0.952	0.943
	CR2	0.825			
	CR3	0.861			
	CR4	0.513			
	CR5	0.844			
	CR6	0.820			
	CR7	0.518			
	CR8	0.805			
	CR9	0.848			
	CR10	0.836			
	CR11	0.851			
	CR12	0.839			
Learning effectiveness (LE)	LE1	0.858	0.743	0.920	0.885
	LE2	0.872			
	LE3	0.864			
	LE4	0.853			

**Table 8 behavsci-15-00560-t008:** Analysis of HTMT discriminant validity.

	ES	LE	CS	CR	TC	NB	VB
ES							
LE	0.320						
CS	0.291	0.439					
CR	0.290	0.572	0.445				
TC	0.320	0.336	0.230	0.349			
NB	0.210	0.304	0.280	0.270	0.205		
VB	0.104	0.344	0.201	0.306	0.206	0.202	

Notes: ES = emotional support, LE = learning effectiveness, CS = communication satisfaction, CR = teacher’s credibility, TC = teaching content, NB = nonverbal behavior, VB = verbal behavior.

**Table 9 behavsci-15-00560-t009:** Assessment of structural model with the bootstrapping procedure.

Path Hypotheses	Sample Mean	Std	*t*-Value	*p*-Value	F-Square	Findings
VB -> CR	0.207	0.045	4.598	0	0.051	Supported
VB -> CS	0.113	0.046	2.452	0.014	0.006	Supported
VB -> LE	0.230	0.026	4.391	0	0.028	Supported
NB -> CR	0.148	0.045	3.276	0.001	0.026	Supported
NB -> CS	0.177	0.047	3.813	0	0.033	Supported
NB -> LE	0.168	0.025	4.117	0	0.031	Supported
ES -> CR	0.162	0.047	3.432	0.001	0.030	Supported
ES -> CS	0.192	0.046	4.209	0	0.038	Supported
ES -> LE	0.187	0.024	4.73	0	0.011	Supported
TC -> CR	0.210	0.044	4.813	0	0.049	Supported
TC -> CS	0.099	0.048	2.085	0.037	0.021	Supported
TC -> LE	0.170	0.025	4.497	0	0.020	Supported
CR -> LE	0.347	0.043	10.334	0	0.137	Supported
CS -> LE	0.152	0.047	4.412	0	0.029	Supported

Notes: ES = emotional support, LE = learning effectiveness, CS = communication satisfaction, CR = teacher’s credibility, TC = teaching content, NB = nonverbal behavior, VB = verbal behavior.

**Table 10 behavsci-15-00560-t010:** Mediating effects of communication satisfaction (CS) and teacher credibility (CR).

	Original Sample (O)	Sample Mean (M)	Standard Deviation (STDEV)	*t* Statistics (|O/STDEV|)	*p* Values	VAF %	Mediation Effect
VB -> CR -> LE	0.056	0.072	0.019	3.77	0	44.72	Yes
VB -> CS -> LE	0.029	0.017	0.009	1.877	0.061	16.03	No
NB -> CR -> LE	0.051	0.053	0.018	2.905	0.004	65.57	Yes
NB -> CS -> LE	0.027	0.028	0.012	2.3	0.022	25.71	Yes
ES -> CR -> LE	0.073	0.057	0.018	3.107	0.002	39.43	Yes
ES -> CS -> LE	0.015	0.03	0.012	2.53	0.011	25.22	Yes
TC -> CR -> LE	0.072	0.074	0.019	3.896	0	45.34	Yes
TC -> CS -> LE	0.017	0.015	0.009	1.751	0.08	14.56	No

Notes: ES = emotional support, LE = learning effectiveness, CS = communication satisfaction, CR = teacher’s credibility, TC = teaching content, NB = nonverbal behavior, VB = verbal behavior.

## Data Availability

The original contributions presented in this study are included in the article. Further inquiries can be directed to the corresponding author.

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
