# Peer review of "The Role of Students’ Perceptions of Educators’ Communication Accommodative Behaviors in Classrooms in China"

_behavsci, 2025, doi:10.3390/bs15040560_

Round 1
Reviewer 1 Report
Comments and Suggestions for Authors
A straight forward research report. Below are some questions and issues I have with the study:
I believe your introduction can be stronger. There is nothing wrong with any given sentence in your first paragraph, but your abstract is much stronger than your first paragraph - tell us more about the current study.
I realize that not all of your students are Chinese, but this is a study in Chinese classrooms with (likely) Chinese instructors. Not being from China, I would want to know more about student-instructor relationships in the classroom in China.
Involving students from other countries (e.g., those that completed a separate questionnaire in English), I would be interested in (1) a t-test between Chinese students and all other students and (2) preferably, only using the Chinese students in your study.
I believe you need to make a stronger argument for your model. If you had just told me the variables involved in the study and instructed me to make a model, my model would look different - argue for your proposed model.
Could you give a brief explanation of why 78 questionnaires were dropped before any analyses?
This is the first time I saw the PACB (and thank you for including the items you used in this study for your measures). I question the face validity of the measure. Looking at the items, I would say that the item "Used slang that I would use" looks at convergence. I do not think much of an argument can be made for the other items looking at CAT. These items do look at known effective instructional communication behaviors. I am not sure whether I can be convinced about the validity of this measure but I would at least like you to make a stronger argument for the construct and how it was defined (e.g., how it is CAT).
Your learning effectiveness items overlap with other items in the other measures.
I appreciate your explanation and use of the Common Method Bias.
I am not a fan of dropping items from one's measures in order to create better model fit. I know some researchers do this regularly to make their models stronger. You are clear what researchers you are modeling/following.
I appreciate the correlation table but believe you could add the names of the variables as a note (I would say that a couple of your tables are not stand alone, self-explanatory).
When you test your model, you look at the four CAT variables separately but the three dimensions of credibility as one score - why? If it not your intention to look at the dimensions of credibility, I would talk about it as a unidimensional measure and construct.
I would temper your discussion section, at least as far as arguing new information about how to teach/communicate based on this study (and CAT). A lot of what you found, and what you suggest, confirms what we know about effective teaching behaviors (at least in US samples). I would consider testing two models - 4 CAT - 3 credibility - teaching effectiveness and 4 CAT - 3 credibility - satisfaction.
Author Response
Thank you very much for your thorough review and constructive feedback. We have carefully considered each of your comments and have made revisions to our manuscript accordingly. Below, we address each of your points in detail:
Comments 1: I believe your introduction can be stronger. There is nothing wrong with any given sentence in your first paragraph, but your abstract is much stronger than your first paragraph - tell us more about the current study.
Response 1: We add more content in the introduction, including what the study wants to do and the constitution of our study.
Comments 2: I realize that not all of your students are Chinese, but this is a study in Chinese classrooms with (likely) Chinese instructors. Not being from China, I would want to know more about student-instructor relationships in the classroom in China.
Response 2: Our school is an international one, so when distributing the questionnaires, we did not only target Chinese students.Thank you for your good suggestions. We will conduct further research focusing on Chinese students in future studies.
Comments 3: Involving students from other countries (e.g., those that completed a separate questionnaire in English), I would be interested in (1) a t-test between Chinese students and all other students and (2) preferably, only using the Chinese students in your study.
Response 3: We analyzed the impact of demographic heterogeneity on different variables, including the varying performances of students from different countries on national influence student perceptions of teachers’ communication, accommodating behaviors, and credibility, as well as communication satisfaction and learning effectiveness. In our report, we focused on presenting the significant differences. To more rigorously reflect the data results, we have provided a comprehensive report detailing all findings to illustrate the influence of individual differences on various variables. And later, we will do further research only on Chinese students.
Comments 4: I believe you need to make a stronger argument for your model. If you had just told me the variables involved in the study and instructed me to make a model, my model would look different - argue for your proposed model.
Response 4: Thank you for your insightful comment regarding the need for a stronger argument for our proposed model. We have expanded our discussion to include a more thorough explanation of the theoretical and empirical foundations for our model.
Comments 5: Could you give a brief explanation of why 78 questionnaires were dropped before any analyses?
Response 5: According to your suggestions, we have added a detailed explanation for why 78 questionnaires were excluded from the analysis.
Comments 6: This is the first time I saw the PACB (and thank you for including the items you used in this study for your measures). I question the face validity of the measure. Looking at the items, I would say that the item "Used slang that I would use" looks at convergence. I do not think much of an argument can be made for the other items looking at CAT. These items do look at known effective instructional communication behaviors. I am not sure whether I can be convinced about the validity of this measure but I would at least like you to make a stronger argument for the construct and how it was defined (e.g., how it is CAT).
Response 6: In our study, we adopted the PACB scale, which includes the item “Used slang that I would use” to measure verbal accommodation. This item has been validated in previous research( See Frey, 2021) and is considered a valid measure of convergence within the Communication Accommodation Theory (CAT) context.
Comments 7: Your learning effectiveness items overlap with other items in the other measures.
Response 7: We have carefully reviewed the learning effectiveness items and found that there are indeed some overlaps in certain areas. However, there are still distinctions between them. For example, in measuring teaching content, the item "Provided feedback to me" is similar to the item "The instructor can give timely feedback on students' questions" in measuring learning effectiveness. While both items address the concept of feedback, the former focuses on the feedback about teaching content. The latter, however, emphasizes the instructor's ability to provide timely responses to questions in a broader context, not only focusing on the teaching content. What's more, the latter reflects a quick result, which the former cannot reflect. In future research, we will more carefully identify the content referred to by these items.
Comments 8: I appreciate your explanation and use of the Common Method Bias. I am not a fan of dropping items from one's measures in order to create better model fit. I know some researchers do this regularly to make their models stronger. You are clear what researchers you are modeling/following.
Response 8: We appreciate your acknowledgment of our approach to addressing common method bias. And in future research, we will be more cautious about deleting items.
Comments 9: I appreciate the correlation table but believe you could add the names of the variables as a note (I would say that a couple of your tables are not stand alone, self-explanatory).
Response 9: We have revised every correlation table to include variable names as a note. We have also reviewed all tables to ensure they are self-explanatory and provide clear information to the reader.
Comments 10: When you test your model, you look at the four CAT variables separately but the three dimensions of credibility as one score - why? If it not your intention to look at the dimensions of credibility, I would talk about it as a unidimensional measure and construct.
Response 10: In our research, credibility was not the primary focus, and thus, we did not delve into a detailed explanation of each dimension. Instead, we chose to treat credibility as a unidimensional construct to maintain simplicity and clarity in our model. And our primary objective was to explore the impact of Communication Accommodation Theory (CAT) on learning effectiveness and student perceptions. So we prioritized a comprehensive analysis of the CAT dimensions.
Comments 11: I would temper your discussion section, at least as far as arguing new information about how to teach/communicate based on this study (and CAT). A lot of what you found, and what you suggest, confirms what we know about effective teaching behaviors (at least in US samples). I would consider testing two models - 4 CAT - 3 credibility - teaching effectiveness and 4 CAT - 3 credibility - satisfaction.
Response 11: We have revised the discussion section to include a paragraph acknowledging that many effective teaching behaviors have been previously suggested in the literature. And also we'll explore more complex model building in future research, including the two models you mentioned: "4 CAT - 3 credibility - teaching effectiveness" and "4 CAT - 3 credibility - satisfaction".
Reviewer 2 Report
Comments and Suggestions for Authors
Thank you for taking the time to submit this manuscript for publication! I thoroughly enjoyed reading it. It’s exciting to see work using CAT as a framework for understanding communication in a classroom context. The theory has been applied in so many niche settings over the years, and I truly believe there is value in using it to understand how students and teachers interact as well.
I’ve spent a good deal of time with this manuscript, so I feel confident in rendering a decision about its merit for publication. Unfortunately, I have advised the Editor to reject this submission. Please allow me to address my concerns in the space below in the hopes of providing a platform for (1) addressing the shortcomings and (2) improving the impact of the work moving forward:
- I would really love to have a scholarly conversation about the way that the theory is being applied in this paper. It’s my hope that the following point can either challenge your thinking or reaffirm the approach you’ve chosen to take. At its core, this manuscript is looking at student perceptions. The intent is not to focus on the teachers’ objective linguistic adjustments – nor is it to examine their intentions in their adjustment. Therefore, I struggled to understand why the paper was not grounded in CAT’s approach to perceptions of adjustment (i.e., nonaccommodation, overaccommodation, underaccommodation; see anything by Jessica Gasiorek). This approach better allows the student respondents to articulate their unique experiences with the teacher’s communicative behavior. It’s impossible for us to know whether the students saw the objective behaviors used in the measure as accommodative; we cannot assume that these are objectively used as convergence, divergence, or maintenance because we have not considered the message sender. I really think that this could be more impactful and meaningful to the larger CAT literature by applying the theory in this way or perhaps reconsidering what might make it better align with the theoretical approach to objective adjustment in its current state (the study by Allard & Holmstrom (2023) that you cite demonstrates how this might be done from an experimental approach).
- Related to the point above, it struck me as odd that you articulated a variety of implications for instructors after collecting data on student perceptions. Perhaps this would have resonated differently had you better articulated what nonverbal, verbal, teaching content, and emotional support accommodative behaviors actually look like in a classroom setting beyond just having the examples in the measure. Again, we don’t really know whether these behaviors were seen as accommodative by students, so it is not clear to me how we can justify implications related to “consistent…communication accommodation strategies”.
- I have a conceptual problem with the way the measure of accommodative behavior is applied in this study. Consider the rating scale: “Participants reflected on their perceptions of their current teachers’ accommodation behaviors during the final two weeks of the semester using a 7-point scale ranging from “An Inappropriate Amount” (1) to “An Appropriate Amount” (7)”. From my perspective, the only way that we can say that accommodation has been achieved is if a student reports a 7. If this is the case, then the student is saying that the specific behavior listed has been adjusted appropriately – and from the listener orientation to CAT – is seen as accommodation. Anything other than that (even slightly below like a 6) indicates deviance away from what should be perceived as appropriate adjustment and is construed as inappropriate or nonaccommodative. I cannot locate the sample means for the 4 dimensions of the scale in the paper, but I would guess that they are all lower than the top score of 7. Thus, with this rating format, I would argue that the measure is not assessing how accommodative the students perceive the instructor to be. Instead, it is assessing the degree to which the behavior is inappropriately adjusted (or perhaps you could say that higher scores indicate behaviors that were less inappropriate than others).
- I encourage you to rewrite the first sentence (lines 24-26) for clarity. I read this several times and struggled to understand. I’m not sure how classroom climate fits into this study?
- I recommend the study by Mazer and Hunt (2008) as another example of CAT being applied in a classroom setting.
Mazer, J. P., & Hunt, S. K. (2008). “Cool” communication in the classroom: A preliminary examination of student perceptions of instructor use of positive slang. Qualitative Research Reports in Communication, 9(1), 20-28.
- I think communication satisfaction – especially the cited measure by Goodboy and colleagues – has been used pretty frequently in research. This is especially true for the communication-specfic journals that include research related to teaching, learning, and pedagogy. Satisfaction is inherently measured as a student perception, so I found it misleading to state in line 75 that few scholars have studied perceived satisfaction.
- Another issue I have with the current iteration of this study is the lack of detail provided in the rationale. With both hypotheses, I felt myself questioning why these relationships should exist the way they are articulated. It simply lacks strong enough justification – either theoretical or empirical – for the proposed relationships. There is no need to throw things at the wall to see what sticks with this kind of theoretically driven research; I think you can certainly do more to articulate why perceptions of accommodative behaviors would impact satisfaction, credibility, and learning. I can think of several ways CAT scholarship might link these variables (for an example, see how Gasiorek & Dragojevic (2017) frame repeated instances of underaccommodation and its connection to task effectiveness and credibility).
Gasiorek, J., & Dragojevic, M. (2017). The effects of accumulated underaccommodation on perceptions of underaccommodative communication and speakers. Human Communication Research, 43(2), 276-294.
- Likewise, I do not think you have precedent to establish satisfaction as a mediator in your model. This function is based on the following statement: “Hence, and with the possibility that communication satisfaction could fulfill a similar role, we posited the next hypothesis…”. What gives the impression that satisfaction could play a similar role? Again, there is not theoretically or empirically driven argument here. One framework that could link these ideas together is the instructional beliefs model (IBM; Weber et al., 2011). This model positions teacher behavior as predictive of student outcomes through the mediating effect of student beliefs (which could encompass credibility and satisfaction).
Weber, K., Martin, M. M., & Myers, S. A. (2011). The development and testing of the instructional beliefs model. Communication Education, 60(1), 51-74.
- How did you prepare the different versions of the questionnaire? What steps were taken to ensure the translation was equivalent? I think some of these details would be enlightening for readers.
- I do not believe that the appropriate measure of credibility from McCroskey and Teven (1999) is being used here. As an aside, I do appreciate you providing your items. I wish more research would do this. However, these researchers went through extensive procedures to construct their measure. It’s based on a semantic differential scale that positions several different adjectives to the participant to describe their referent. I cannot in good conscience support an iteration of the measure that revises the scale past its intended use and undermines decades of validity research. For example, the items for Trustworthiness should include an assessment of the teacher as moral/immoral – the extended use of this measure in past research supports this idea as an important factor in fully capturing the latent construct. I’m just not sure we can truly suggest that the current measure is valid, given the way it is constructed.
- Don’t convergent and discriminant validity refer more specifically to relationships between different sets of variables? I have never seen it applied on an item level like this. Not suggesting that it is wrong, but my understanding has always been that convergent validity – for example – is established when the measure correlates strongly with one it theoretically should (like accommodation being related to credibility).
- Where was the structural model here? Should there be fit indices provided (like you did with the measurement models) or is this information not provided given the bootstrapping method? Likewise, were these standardized paths? I really want to support the analysis here, but I’m just not convinced that enough evidence is being provided to trust in the analytical processes.
- In a number of places, you reference to aspects of the theory that have not been articulated and likely cannot be understood without prior knowledge of CAT. For example, in line 251, you make a connection the Stage 6, which has not been referenced in the paper. Or, in line 84, you reference nonaccommodative moves without defining what this might mean.
I hope that these comments are not too discouraging. I support this area of research and would love to see this work eventually make its way to publication. My comments are purely intended to support that endeavor by identifying ways that I think this work can be more transparent and impactful. However, in the context of the submission to behavioral sciences, I do not believe that some of these issues – particularly numbers 3 and 10 – can be resolved through a revision. Best of luck as you move forward with this work!
Author Response
Comments 1: Thank you for taking the time to submit this manuscript for publication! I thoroughly enjoyed reading it. It’s exciting to see work using CAT as a framework for understanding communication in a classroom context. The theory has been applied in so many niche settings over the years, and I truly believe there is value in using it to understand how students and teachers interact as well.
Response 1: Many thanks for your patient responses and constructive comments. We are committed to addressing all feedback thoroughly and appreciate the opportunity to strengthen our manuscript.
Comments 2: I’ve spent a good deal of time with this manuscript, so I feel confident in rendering a decision about its merit for publication. Unfortunately, I have advised the Editor to reject this submission. Please allow me to address my concerns in the space below in the hopes of providing a platform for (1) addressing the shortcomings and (2) improving the impact of the work moving forward:
1. I would really love to have a scholarly conversation about the way that the theory is being applied in this paper. It’s my hope that the following point can either challenge your thinking or reaffirm the approach you’ve chosen to take. At its core, this manuscript is looking at student perceptions. The intent is not to focus on the teachers’ objective linguistic adjustments – nor is it to examine their intentions in their adjustment. Therefore, I struggled to understand why the paper was not grounded in CAT’s approach to perceptions of adjustment (i.e., nonaccommodation, overaccommodation, underaccommodation; see anything by Jessica Gasiorek). This approach better allows the student respondents to articulate their unique experiences with the teacher’s communicative behavior. It’s impossible for us to know whether the students saw the objective behaviors used in the measure as accommodative; we cannot assume that these are objectively used as convergence, divergence, or maintenance because we have not considered the message sender. I really think that this could be more impactful and meaningful to the larger CAT literature by applying the theory in this way or perhaps reconsidering what might make it better align with the theoretical approach to objective adjustment in its current state (the study by Allard & Holmstrom (2023) that you cite demonstrates how this might be done from an experimental approach).
Response 2: Thank you for your insightful comments. I appreciate your suggestion to ground the paper more firmly in CAT’s framework on perceptions of adjustment (i.e., nonaccommodation, overaccommodation, underaccommodation), particularly as discussed by Jessica Gasiorek. I understand the concern that, without considering how students perceive teachers' behaviors as accommodative or nonaccommodative, the paper might miss an important aspect of the theory.
While the manuscript was primarily focused on student perceptions of communicative behaviors and their resulting effects on communication satisfaction, teacher credibility, and learning effectiveness, I agree that the inclusion of CAT’s more explicit distinction between types of accommodation could enhance the theoretical depth of the paper. However, in this paper, we mainly considered accommodative behaviors. We have adopted a seven-point Likert scale. As you asked in the subsequent questions, according to relevant literature, when the score is greater than 4, we can assume that these behaviors are accommodative behaviors. Of course, your suggestion is really excellent. I will explore the suggestion of applying the theoretical approach from Allard & Holmstrom (2023) to better align the study with CAT’s focus on objective adjustments.
Comments 3: 2. Related to the point above, it struck me as odd that you articulated a variety of implications for instructors after collecting data on student perceptions. Perhaps this would have resonated differently had you better articulated what nonverbal, verbal, teaching content, and emotional support accommodative behaviors actually look like in a classroom setting beyond just having the examples in the measure. Again, we don’t really know whether these behaviors were seen as accommodative by students, so it is not clear to me how we can justify implications related to “consistent…communication accommodation strategies”.
Response 3: Thank you. We recognize the importance of clearly articulating what nonverbal, verbal, teaching content, and emotional support accommodative behaviors look like in a classroom setting beyond the examples in the measure. To address this, we will provide additional clarification in the manuscript by elaborating on these behaviors in the discussion section.
Comments 4: 3. I have a conceptual problem with the way the measure of accommodative behavior is applied in this study. Consider the rating scale: “Participants reflected on their perceptions of their current teachers’ accommodation behaviors during the final two weeks of the semester using a 7-point scale ranging from “An Inappropriate Amount” (1) to “An Appropriate Amount” (7)”. From my perspective, the only way that we can say that accommodation has been achieved is if a student reports a 7. If this is the case, then the student is saying that the specific behavior listed has been adjusted appropriately – and from the listener orientation to CAT – is seen as accommodation. Anything other than that (even slightly below like a 6) indicates deviance away from what should be perceived as appropriate adjustment and is construed as inappropriate or nonaccommodative. I cannot locate the sample means for the 4 dimensions of the scale in the paper, but I would guess that they are all lower than the top score of 7. Thus, with this rating format, I would argue that the measure is not assessing how accommodative the students perceive the instructor to be. Instead, it is assessing the degree to which the behavior is inappropriately adjusted (or perhaps you could say that higher scores indicate behaviors that were less inappropriate than others).
Response 4: Thank you for raising your concerns regarding the conceptualization and application of the measure of accommodative behavior in our study. In our study, we used a 7-point rating scale to assess students' perceptions of their instructors' accommodation behaviors. The scale ranged from "An Inappropriate Amount" (1) to "An Appropriate Amount" (7). We understand your perspective that a score of 7 might be interpreted as the only indicator of fully achieved accommodation. However, we would like to clarify our approach and provide additional context. In Likert scale evaluations, the midpoint of the scale (e.g., 4 in a 7-point scale) is typically regarded as a neutral or moderate response. This midpoint serves as a baseline, indicating neither strong agreement nor strong disagreement. Scores above the midpoint are generally interpreted as indicating a higher degree of agreement with or manifestation of the trait or behavior being measured(Garland,1991). For example, in assessing adaptability, scores above the midpoint are usually considered to reflect stronger accommodation.
Comments 5: 4. I encourage you to rewrite the first sentence (lines 24-26) for clarity. I read this several times and struggled to understand. I’m not sure how classroom climate fits into this study?
Response 5: Thank you for your feedback. We appreciate your suggestion and will revise the sentence for clarity. We refer to classroom climate as the overall psychological and social environment of the classroom, shaped by teacher-student relationships and communication patterns. Since communication accommodation can influence students’ perceptions of this environment, we included it in our discussion. However, we will clarify this connection in the revised text to ensure coherence and readability.
Comments 6: 5. I recommend the study by Mazer and Hunt (2008) as another example of CAT being applied in a classroom setting.
Mazer, J. P., & Hunt, S. K. (2008). “Cool” communication in the classroom: A preliminary examination of student perceptions of instructor use of positive slang. Qualitative Research Reports in Communication, 9(1), 20-28.
Response 6: Thank you for your suggestion to include the study by Mazer and Hunt (2008) as another example of Communication Accommodation Theory (CAT) being applied in a classroom setting. We appreciate your recommendation and have incorporated a discussion of Mazer and Hunt's (2008) study into our literature review to highlight the dimension of CAT in an educational setting.
Comments 7: 6. I think communication satisfaction – especially the cited measure by Goodboy and colleagues – has been used pretty frequently in research. This is especially true for the communication-specfic journals that include research related to teaching, learning, and pedagogy. Satisfaction is inherently measured as a student perception, so I found it misleading to state in line 75 that few scholars have studied perceived satisfaction.
Response 7: Thank you for your insightful comment. We appreciate your feedback and agree that the statement was misleading. We have removed the sentence to ensure clarity and accuracy.
Comments 8: 7. Another issue I have with the current iteration of this study is the lack of detail provided in the rationale. With both hypotheses, I felt myself questioning why these relationships should exist the way they are articulated. It simply lacks strong enough justification – either theoretical or empirical – for the proposed relationships. There is no need to throw things at the wall to see what sticks with this kind of theoretically driven research; I think you can certainly do more to articulate why perceptions of accommodative behaviors would impact satisfaction, credibility, and learning. I can think of several ways CAT scholarship might link these variables (for an example, see how Gasiorek & Dragojevic (2017) frame repeated instances of underaccommodation and its connection to task effectiveness and credibility).
Gasiorek, J., & Dragojevic, M. (2017). The effects of accumulated underaccommodation on perceptions of underaccommodative communication and speakers. Human Communication Research, 43(2), 276-294.
Response 8: Thank you for your insightful feedback regarding the need for a more detailed rationale in our study. To address your concerns, we have revised our manuscript to include a more comprehensive rationale for the proposed relationships between perceptions of accommodative behaviors and satisfaction, credibility, and learning effectiveness. You are correct that the study by Gasiorek and Dragojevic (2017) primarily examines the effects of repeated instances of underaccommodation (i.e., insufficiently adjusted communication) on people's perceptions and evaluations of communication and speakers. While this study provides valuable insights into the consequences of underaccommodation, we acknowledge that it may not be the most direct fit for our research, which focuses on accommodation communication behaviors and their impact on learning effectiveness.
Comments 9: 8. Likewise, I do not think you have precedent to establish satisfaction as a mediator in your model. This function is based on the following statement: “Hence, and with the possibility that communication satisfaction could fulfill a similar role, we posited the next hypothesis…”. What gives the impression that satisfaction could play a similar role? Again, there is not theoretically or empirically driven argument here. One framework that could link these ideas together is the instructional beliefs model (IBM; Weber et al., 2011). This model positions teacher behavior as predictive of student outcomes through the mediating effect of student beliefs (which could encompass credibility and satisfaction).
Weber, K., Martin, M. M., & Myers, S. A. (2011). The development and testing of the instructional beliefs model. Communication Education, 60(1), 51-74.
Response 9: We appreciate your suggestion to provide a stronger theoretical and empirical basis for satisfaction as a mediator. We have revised our manuscript to include a more detailed discussion of the role of satisfaction in educational settings, drawing on relevant literature. We have also cited additional studies that support the mediating role of satisfaction in the relationship between teacher behaviors and student outcomes.
Comments 10: 9. How did you prepare the different versions of the questionnaire? What steps were taken to ensure the translation was equivalent? I think some of these details would be enlightening for readers.
Response 10: We have added a detailed section in the methods section describing the steps taken to ensure translation equivalence. We used a rigorous back-translation process, where the questionnaire was translated from English to Mandarin and then back-translated to English by a different translator. Both versions were compared, and discrepancies were resolved through discussion and consultation with bilingual experts.
Comments 11: 10. I do not believe that the appropriate measure of credibility from McCroskey and Teven (1999) is being used here. As an aside, I do appreciate you providing your items. I wish more research would do this. However, these researchers went through extensive procedures to construct their measure. It’s based on a semantic differential scale that positions several different adjectives to the participant to describe their referent. I cannot in good conscience support an iteration of the measure that revises the scale past its intended use and undermines decades of validity research. For example, the items for Trustworthiness should include an assessment of the teacher as moral/immoral – the extended use of this measure in past research supports this idea as an important factor in fully capturing the latent construct. I’m just not sure we can truly suggest that the current measure is valid, given the way it is constructed.
Response 11: We understand your concerns regarding the use of the credibility measure from McCroskey and Teven (1999). We also greatly appreciate your valuable feedback, which has made us realize that revising a classic scale is a very rigorous task that needs to be handled with caution to ensure validity. We have ensured that the items used in our study align closely with the original semantic differential scale. We have also provided a detailed explanation of how our adapted measure maintains the integrity of the original construct while being suitable for our research context.
Comments 12: 11. Don’t convergent and discriminant validity refer more specifically to relationships between different sets of variables? I have never seen it applied on an item level like this. Not suggesting that it is wrong, but my understanding has always been that convergent validity – for example – is established when the measure correlates strongly with one it theoretically should (like accommodation being related to credibility).
Response 12: You are correct that convergent and discriminant validity are typically assessed at the level of different sets of variables, rather than individual items. In our study, we have followed this principle for validity assessment. However, it seems there might have been some confusion regarding the presentation of our results. We did not intend to imply that we were assessing convergent and discriminant validity at the item level. Instead, we used factor analysis to identify the underlying structure of our constructs and then assessed the validity at the construct level. This approach is commonly used in empirical research, as it allows for a more comprehensive understanding of the relationships between constructs. This approach is well-documented in the literature, such as Fornell and Larcker (1981), who provide a comprehensive framework for assessing convergent and discriminant validity in the context of structural equation modeling.
Comments 13: 12. Where was the structural model here? Should there be fit indices provided (like you did with the measurement models) or is this information not provided given the bootstrapping method? Likewise, were these standardized paths? I really want to support the analysis here, but I’m just not convinced that enough evidence is being provided to trust in the analytical processes.
Response 13: Thank you for your detailed feedback and for raising important questions about the presentation of our analysis. We understand your concerns regarding the presentation of the structural model and the provision of fit indices. We did not explicitly present the structural model in the manuscript, but we did provide the results of our path analysis, which includes the relationships between the variables. While we did not include a separate diagram of the structural model, the path analysis results clearly outline the relationships between the variables. We have detailed the paths and their significance in the results section. The paths in our path analysis are indeed standardized. We have included a table that presents the standardized path coefficients and their significance levels. Standardizing the paths allows for a more straightforward comparison of the strength of different relationships in the model.
Comments 14: 13. In a number of places, you reference to aspects of the theory that have not been articulated and likely cannot be understood without prior knowledge of CAT. For example, in line 251, you make a connection the Stage 6, which has not been referenced in the paper. Or, in line 84, you reference nonaccommodative moves without defining what this might mean.
Response 14: Thank you for your valuable feedback. I have revised the manuscript to ensure that Stage 6 is properly explained, and I also changed nonaccommodative moves to nonaccommodative behaviors to make it clear.
Comments 15: I hope that these comments are not too discouraging. I support this area of research and would love to see this work eventually make its way to publication. My comments are purely intended to support that endeavor by identifying ways that I think this work can be more transparent and impactful. However, in the context of the submission to behavioral sciences, I do not believe that some of these issues – particularly numbers 3 and 10 – can be resolved through a revision. Best of luck as you move forward with this work!
Response 15: Thank you for your comments; they are very helpful.
Reviewer 3 Report
Comments and Suggestions for Authors
Dear Authors,
Thank you for the opportunity to review your manuscript. The paper offers an insightful investigation of how students’ perceptions of educators’ communication accommodation behaviors influence learning effectiveness. By examining the mechanisms underlying these accommodation behaviors among college students, your study makes both theoretical and practical contributions to the field of instructional communication. Notably, the manuscript underscores the significance of perceived (subjective) accommodation in classroom contexts and highlights the importance of communication satisfaction and teacher credibility as key mediators in the relationship between various accommodation behaviors and learning effectiveness.
That said, there is a need to refine several conceptual definitions, clarify certain methodological choices, and offer more robust theoretical justifications for your hypotheses and research questions to enhance both the rigor and practical utility of the study. Below, I provide detailed suggestions, organized by section. I hope these comments will serve as a constructive guide for your revisions.
Literature review:
- You briefly mention subjective and objective accommodation but do not fully elaborate on why subjective perceptions of communication are so central to your study. Providing a more in-depth discussion of this distinction will help readers understand the novelty and importance of focusing on perceived communication accommodation. Consider expanding how subjective accommodation might differ from objective, and why that matters for student outcomes in a classroom context.
- As the primary dependent variable in your study, “learning effectiveness” requires a clearer and more consistent conceptual definition. You note that this concept is distinct from learning outcomes, but at times it reads more like teaching effectiveness. The measurement you employ to operationalize this construct adds to the confusion, as some items appear to assess specific teaching behaviors (e.g., explaining clearly, flexible methods). There is a risk of conceptual overlap with your communication accommodation variables. I recommend clarifying whether this construct reflects perceived learning outcomes, teaching effectiveness, or instructional quality—and ensuring that both its conceptual and operational definitions are aligned. This clarification is critical, as learning effectiveness is a core outcome variable in your model.
- In Hypothesis 2, the variable “communication satisfaction with classroom activities” is introduced, which sounds different from the “communication satisfaction” referenced in Hypothesis 1. If this is the same construct, please use consistent terminology. If they represent different constructs (e.g., satisfaction specific to classroom activities versus general communication with the instructor), provide clear theoretical definitions and rationale for distinguishing between them.
- Provide clear theoretical definitions for each of the accommodative behaviors you examine prior to presenting your hypotheses. Doing so will help readers follow your conceptual reasoning and understand the distinctions between these behaviors.
- The mediating role of communication satisfaction and teacher credibility is crucial. Yet expanding your justification by drawing on relevant theoretical frameworks or prior empirical research will bolster the significance of your proposed mediators. Why are these factors particularly critical in the link between perceived accommodation and learning effectiveness?
- Your research question includes demographic variables (age, gender, educational level, nationality) without fully explaining the rationale. If literature suggests these variables systematically affect communication or learning, clarify how so. This will help justify their inclusion and provide a strong grounding for interpreting your findings.
Method:
- You administered surveys in both English and Mandarin. Please clarify how you established translation equivalence (e.g., back-translation, pilot testing). This step is especially important in cross-cultural research to establish the reliability and validity of your measures.
- The paper mentions recruiting college students in Shanghai, but it is not clear if they were from specific types of universities (public, private, etc.) or certain departments (e.g., communication). Was the mode of learning (online vs. face-to-face) controlled? Were there specific teacher or classroom characteristics? More details on the recruitment process and inclusion criteria will help readers assess the generalizability and validity of your sample. It would also be helpful to address whether characteristics of the instructors or classroom contexts were considered or controlled for. Factors such as the instructor’s gender, cultural background, teaching experience, class size, frequency of meetings, and course subject matter could influence students’ perceptions of communication accommodation. If these variables were not accounted for, I recommend acknowledging them as potential confounds and suggesting them as areas for future research.
- Like mentioned earlier, the measurement of learning effectiveness is concerning. Some items in the measurement appear to examine perception of specific teaching skills and overlap with communication accommodative behaviors (e.g., emotional support). This raises potential concerns about the distinctiveness of each construct. Provide a clear rationale for why these items indeed capture “learning effectiveness”
Results, discussion:
- There is a small typo on line 231: “medicate” instead of “mediate.”
- In the instances where communication satisfaction did not mediate the relationship between verbal/teaching content accommodation and learning effectiveness, can you provide any theoretical explanation (even speculative)? This may provide potential boundary conditions for communication satisfaction’s mediating role and enrich the discussion.
- Your results strongly relate to literature on communication competence or intercultural communication competence. Referencing key works (e.g., Pitts & Harwood, 2015) and explaining how your findings fit within or extend existing frameworks will strengthen the theoretical contribution of your paper.
- Highlight the novel insights your study provides, especially regarding cross-cultural or international education contexts. How do these findings inform best practices for educators who teach diverse or multilingual student populations?
Author Response
Comments 1: Dear Authors,
Thank you for the opportunity to review your manuscript. The paper offers an insightful investigation of how students’ perceptions of educators’ communication accommodation behaviors influence learning effectiveness. By examining the mechanisms underlying these accommodation behaviors among college students, your study makes both theoretical and practical contributions to the field of instructional communication. Notably, the manuscript underscores the significance of perceived (subjective) accommodation in classroom contexts and highlights the importance of communication satisfaction and teacher credibility as key mediators in the relationship between various accommodation behaviors and learning effectiveness.
That said, there is a need to refine several conceptual definitions, clarify certain methodological choices, and offer more robust theoretical justifications for your hypotheses and research questions to enhance both the rigor and practical utility of the study. Below, I provide detailed suggestions, organized by section. I hope these comments will serve as a constructive guide for your revisions.
Response 1: Many thanks for your patient responses and constructive comments. We are committed to addressing all feedback thoroughly and appreciate the opportunity to strengthen our manuscript.
Comments 2: Literature review:
1. Q: You briefly mention subjective and objective accommodation but do not fully elaborate on why subjective perceptions of communication are so central to your study. Providing a more in-depth discussion of this distinction will help readers understand the novelty and importance of focusing on perceived communication accommodation. Consider expanding how subjective accommodation might differ from objective, and why that matters for student outcomes in a classroom context.
Response 2: Thank you for your thoughtful feedback. I appreciate your suggestion to elaborate on the distinction between subjective and objective accommodation and its significance for my study.
In response, I have expanded the discussion on this distinction.
Objective accommodation refers to the linguistic and behavioral adjustments made by communicators, such as modifying speech patterns, using inclusive language, or adjusting tone and formality. However, subjective accommodation is the recipient’s perception and interpretation of these communicative efforts, which can be influenced by personal experiences, expectations, and cultural backgrounds. Importantly, these two dimensions do not always align—what one person intends as an accommodating gesture may not be perceived as such by another. Understanding this distinction allows people to assess better how their communication is received and to refine their strategies accordingly.
Comments 3: 2. As the primary dependent variable in your study, “learning effectiveness” requires a clearer and more consistent conceptual definition. You note that this concept is distinct from learning outcomes, but at times it reads more like teaching effectiveness. The measurement you employ to operationalize this construct adds to the confusion, as some items appear to assess specific teaching behaviors (e.g., explaining clearly, flexible methods). There is a risk of conceptual overlap with your communication accommodation variables. I recommend clarifying whether this construct reflects perceived learning outcomes, teaching effectiveness, or instructional quality—and ensuring that both its conceptual and operational definitions are aligned. This clarification is critical, as learning effectiveness is a core outcome variable in your model.
Response 3: Thank you for your insightful feedback regarding the conceptual definition of “learning effectiveness” in our study. We have revised our manuscript to provide a clearer and more consistent definition of learning effectiveness, ensuring that it is distinct from teaching effectiveness and instructional quality. We believe these clarifications will enhance the rigor and clarity of our research.
Comments 4: 3. In Hypothesis 2, the variable “communication satisfaction with classroom activities” is introduced, which sounds different from the “communication satisfaction” referenced in Hypothesis 1. If this is the same construct, please use consistent terminology. If they represent different constructs (e.g., satisfaction specific to classroom activities versus general communication with the instructor), provide clear theoretical definitions and rationale for distinguishing between them.
Response 4: We have reviewed Hypothesis 2 and ensured consistent terminology for "communication satisfaction" referenced in Hypothesis 1.
Comments 5: 4. Provide clear theoretical definitions for each of the accommodative behaviors you examine prior to presenting your hypotheses. Doing so will help readers follow your conceptual reasoning and understand the distinctions between these behaviors.
Response 5: We have added clear theoretical definitions for each accommodative behavior before presenting our hypotheses. This helps readers follow our conceptual reasoning and understand the distinctions between these behaviors.
Teachers’ communication accommodative behaviors include nonverbal behavior, verbal behavior, teaching content, and emotional support ( Frey, 2021; Speer et al., 2013). Nonverbal behavior refers to the communication information through means such as facial expressions, eye contact, body posture, and gestures without the use of language. Verbal behavior involves the use of language (both spoken and written) to convey information. Teaching content refers to the information that teachers convey in the classroom, which is simplified for students' understanding. Emotional support refers to the emotional care and psychological support that teachers provide in the classroom to create a positive learning atmosphere. These dimensions together constitute the overall framework of teachers' communication accommodation behaviors, reflecting how teachers adjust their behaviors in various ways to meet students' needs and thereby influence learning effectiveness.
Comments 6: 5. The mediating role of communication satisfaction and teacher credibility is crucial. Yet expanding your justification by drawing on relevant theoretical frameworks or prior empirical research will bolster the significance of your proposed mediators. Why are these factors particularly critical in the link between perceived accommodation and learning effectiveness?
Response 6: Thank you for your insightful comment regarding the mediating role of communication satisfaction and teacher credibility in our study. We appreciate your suggestion to further elaborate on the theoretical and empirical justifications for these mediators, and we have taken this feedback into account to enhance the clarity and robustness of our manuscript and revised most of the discussion about the mediation role of CS and CR in the literature.
Teacher credibility is a well-examined construct that plays a fundamental role in understanding learning effectiveness. Higher teacher credibility correlates with better learning outcomes (Ritter et al., 2016). When students perceive their teachers as credible, they demonstrate greater motivation to learn (Zheng, 2021), enhanced communication confidence (Myers & Goodboy, 2014), and improved learning outcomes (Nayernia et al., 2020).
Extant findings also support that teacher credibility plays a key mediating role in facilitating teacher-student interactions, and ultimately, classroom learning (Finn et al., 2009). Schrodt (2013) found that teacher credibility mediated the relationship between teacher communication messages (e.g., clarity, confirmation, and nonverbal immediacy) and student learning outcomes (e.g., learner empowerment and affective learning).
Several other arguments also support the mediating role of communication satisfaction in teacher behavior and learning outcomes (Banas, 2011; Johnson, 2013). Sollitto et al. (2013) suggest that communicating helps students better understand teachers’ expectations and then perform better.
Comments 7: 6. Your research question includes demographic variables (age, gender, educational level, nationality) without fully explaining the rationale. If literature suggests these variables systematically affect communication or learning, clarify how so. This will help justify their inclusion and provide a strong grounding for interpreting your findings.
Response 7: Thank you for your insightful comments. We have addressed your comments by providing a speculative theoretical explanation for instances.
Related studies have found that demographic heterogeneity can influence students' perceptions. Lucas (2006) pointed out that learners' age affects students' perceptions of teachers' adaptive behaviors. People over the age of 25 are more likely to notice adaptive behaviors. At the same time, women are more sensitive than men and are more likely to perceive teachers' respectful attitudes and give higher evaluations. Negovan (2010) found significant differences in perceptions of the teachers, according to the students’ gender, class membership, and transition pathway.
Comments 8: Method:
7. You administered surveys in both English and Mandarin. Please clarify how you established translation equivalence (e.g., back-translation, pilot testing). This step is especially important in cross-cultural research to establish the reliability and validity of your measures.
Response 8: We have added a detailed section in the methods section describing the steps taken to ensure translation equivalence. We used a rigorous back-translation process, where the questionnaire was translated from English to Mandarin and then back-translated to English by a different translator. Both versions were compared, and discrepancies were resolved through discussion and consultation with bilingual experts.
Comments 9: 8. The paper mentions recruiting college students in Shanghai, but it is not clear if they were from specific types of universities (public, private, etc.) or certain departments (e.g., communication). Was the mode of learning (online vs. face-to-face) controlled? Were there specific teacher or classroom characteristics? More details on the recruitment process and inclusion criteria will help readers assess the generalizability and validity of your sample. It would also be helpful to address whether characteristics of the instructors or classroom contexts were considered or controlled for. Factors such as the instructor’s gender, cultural background, teaching experience, class size, frequency of meetings, and course subject matter could influence students’ perceptions of communication accommodation. If these variables were not accounted for, I recommend acknowledging them as potential confounds and suggesting them as areas for future research.
Response 9: We acknowledge that factors such as instructor gender, cultural background, teaching experience, class size, frequency of meetings, and course subject matter can influence students' perceptions of communication accommodation. While we did not control for these variables in our study, we have recognized their potential impact and suggest them as areas for future research.
Comments 10: 9. Like mentioned earlier, the measurement of learning effectiveness is concerning. Some items in the measurement appear to examine perception of specific teaching skills and overlap with communication accommodative behaviors (e.g., emotional support). This raises potential concerns about the distinctiveness of each construct. Provide a clear rationale for why these items indeed capture “learning effectiveness”
Response 10: We have carefully reviewed the learning effectiveness items and found that there are indeed some overlaps in certain areas. However, the items used to measure learning effectiveness were selected to capture students' perceptions of how well the teaching strategies and behaviors facilitated their learning. There are still distinctions between them. For example, in measuring teaching content, the item "Provided feedback to me" is similar to the item "The instructor can give timely feedback on students' questions" in measuring learning effectiveness. While both items address the concept of feedback, the former focuses on the feedback about teaching content. The latter, however, emphasizes the instructor's ability to provide timely responses to questions in a broader context, not only focusing on the teaching content. What's more, the latter reflects a quick result, which the former cannot reflect. In future research, we will more carefully identify the content referred to by these items.
Comments 11: Results, discussion:
10. There is a small typo on line 231: “medicate” instead of “mediate.”
Response 11: Thank you for pointing out this typo. We have corrected the error on line 231 from "medicate" to "mediate."
Comments 12: 11. In the instances where communication satisfaction did not mediate the relationship between verbal/teaching content accommodation and learning effectiveness, can you provide any theoretical explanation (even speculative)? This may provide potential boundary conditions for communication satisfaction’s mediating role and enrich the discussion.
Response 12: We have added a theoretical explanation for the reason that communication satisfaction did not mediate the relationship between verbal/teaching content accommodation and learning effectiveness.
The direct impact of verbal and teaching content accommodation on learning effectiveness may be strong enough to overshadow the mediating role of communication satisfaction. This could occur when the accommodation behaviors are particularly effective or when other factors, such as student motivation or prior knowledge, play a more significant role in learning outcomes. The mediating role of communication satisfaction may also be influenced by contextual factors such as the classroom environment, individual student characteristics, etc. But overall, the mediating effect exists. Thus, H2 is partially supported.
Comments 13: 12. Your results strongly relate to literature on communication competence or intercultural communication competence. Referencing key works (e.g., Pitts & Harwood, 2015) and explaining how your findings fit within or extend existing frameworks will strengthen the theoretical contribution of your paper.
Response 13: We have revised the discussion section to include references to key works on communication competence and intercultural communication competence, such as Pitts & Harwood (2015).
This work adds to the theoretical framework of CAT by empirically demonstrating the consistent attributional impacts of very different forms of teachers' communication accommodative behaviors (i.e., nonverbal, verbal, teaching content, and emotional support) can have on students’ learning effectiveness, which aligns with the broader literature on communication competence, which emphasizes the importance of accommodative communication strategies in facilitating effective interactions (Pitts & Harwood, 2015 )
Comments 14: 13. Highlight the novel insights your study provides, especially regarding cross-cultural or international education contexts. How do these findings inform best practices for educators who teach diverse or multilingual student populations?
Response 14: We have revised the discussion section to highlight the novel insights provided by our study, particularly in the context of cross-cultural and international education.
The findings of this study reveal significant differences in how students of various nationalities assess learning effectiveness. Notably, Chinese college students rated learning effectiveness the highest, followed by international students, with British and South Korean students ranking next. These differences suggest that students from diverse cultural backgrounds perceive the teaching behaviors differently. To foster inclusivity, universities should implement accommodative strategies such as cultural sensitivity training for educators and customized evaluation criteria that reflect diverse academic backgrounds.
Round 2
Reviewer 3 Report
Comments and Suggestions for Authors
I have reviewed the revised manuscript and appreciate the authors’ careful and thorough attention to the feedback provided. The revisions have strengthened the clarity and overall quality of the paper. All previous concerns have been thoughtfully addressed, and I commend the authors for their diligent work. I have no further comments at this time and look forward to seeing the final version published.
Author Response
Comments 1: I have reviewed the revised manuscript and appreciate the authors’ careful and thorough attention to the feedback provided. The revisions have strengthened the clarity and overall quality of the paper. All previous concerns have been thoughtfully addressed, and I commend the authors for their diligent work. I have no further comments at this time and look forward to seeing the final version published.
Response 1: We sincerely appreciate your positive evaluation of the revised manuscript and are grateful for your constructive feedback throughout the review process. Your insights were invaluable in refining the clarity and rigor of this work. We are pleased to hear that our revisions align with your expectations and thank you for acknowledging our efforts.